# Design, Synthesis, and Biological Evaluation of Phenyloxadiazole Sulfoxide Derivatives as Potent *Pseudomonas aeruginosa* Biofilm Inhibitors

**DOI:** 10.3390/molecules28093879

**Published:** 2023-05-04

**Authors:** Xinyi Ye, Shen Mao, Yasheng Li, Zhikun Yang, Aoqi Du, Hong Wang

**Affiliations:** Key Laboratory of Marine Fishery Resources Exploitment & Utilization of Zhejiang Province, College of Pharmaceutical Science & Collaborative Innovation Center of Yangtze River Delta Region Green Pharmaceuticals, Zhejiang University of Technology, Hangzhou 310014, China; 1111923009@zjut.edu.cn (S.M.); liyasheng@ahmu.edu.cn (Y.L.); yangzk@zjut.edu.cn (Z.Y.); 2112007103@zjut.edu.cn (A.D.)

**Keywords:** *Pseudomonas aeruginosa*, quorum sensing, inhibitor, biofilm, sulfoxide

## Abstract

With the development of antimicrobial agents, researchers have developed new strategies through key regulatory systems to block the expression of virulence genes without affecting bacterial growth. This strategy can minimize the selective pressure that leads to the emergence of resistance. Quorum sensing (QS) is an intercellular communication system that plays a key role in the regulation of bacterial virulence and biofilm formation. Studies have revealed that the QS system controls 4–6% of the total number of *P. aeruginosa* genes, and quorum sensing inhibitors (QSIs) could be a promising target for developing new prevention and treatment strategies against *P. aeruginosa* infection. In this study, four series of phenyloxadiazole and phenyltetrazole sulfoxide derivatives were synthesized and evaluated for their inhibitory effects on *P. aeruginosa* PAO1 biofilm formation. Our results showed that **5b** had biofilm inhibitory activity and reduced the production of QS-regulated virulence factors in *P. aeruginosa*. In addition, silico molecular docking studies have shown that **5b** binds to the *P. aeruginosa* QS receptor protein LasR through hydrogen bond interaction. Preliminary structure–activity relationship and docking studies show that **5b** has broad application prospects as an anti-biofilm compound, and further research will be carried out in the future to solve the problem of microbial resistance.

## 1. Introduction

As a common Gram-negative bacterium, *Pseudomonas aeruginosa* (*P. aeruginosa* or PA) is one of the main pathogens causing nosocomial infections and ventilator-associated pneumonia. It mainly affects patients that are immunocompromised, such as those with severe burns, chemotherapy, or HIV etc. [1,2,3] The ability of *P. aeruginosa* to form antibiotic-resistant biofilms is a significant factor for its notorious persistence in clinical settings [4]. It has been estimated that over 80% of hospital-acquired infections are biofilm-mediated, and that treatment of these biofilm-based infections costs more than USD 1 billion annually [5,6]. The biofilms’ surface-associated bacteria are microbial communities that attach to inert or living surfaces and encase them in a self-produced extracellular polymeric matrix [7]. Biofilm matrices usually contain polysaccharides, water, proteins, and extracellular DNA, but their composition varies according to bacterial species and environmental conditions [8,9,10]. The bacteria in biofilms are more tolerant to conventional antibiotics, and they are protected from environmental stress and host immune response [11]. *P. aeruginosa* forms biofilm on many surfaces, including cystic fibrosis-affected tissues in lungs and on medicinal implants, such as catheters, artificial hips, and contact lenses, which can cause severe infections that are difficult to treat [6,12,13]. Therefore, PA is an important model to study the growth and inhibition of bacterial biofilms.

As an effective means of evading host defenses, *P. aeruginosa* regulates biofilm formation and the release of virulence factors by a cell–cell communication system termed quorum sensing (QS) [14,15]. There are three major QS systems: *P. aeruginosa*, *las*, and *rhl*, *pqs*. In the *las* system, AHL synthase LasI encoded by the *LasI* gene directs the synthesis of the *N*-(3-oxododecanoyl)-*L*-homoserine lactone (3-oxo-C12-HSL, OdDHL) signal molecule, which binds to the LuxR type transcriptional regulator LasR and activates a series of virulence genes such as LasB elastase, LasA protease, and alkaline protease isogenic expression [15,16,17]. In the *rhl* system, the production of *rhlI*, Rhl directs the synthesis of *N*-butanoyl-*L*-homoserine lactone (C4-HSL, BHL), which forms a complex with the receptor protein RhlR. The complex activates the transcription of the *rhlAB* (rhamnolipid synthesis gene) and the *rhlI* gene itself [18,19]. In addition to the two AHL QS systems in *P. aeruginosa*, a third system acts via the 2-alkyl-4-quinolones autoinducer. The gene product encoded by the *pqsABCD* operon synthesizes 2-heptyl-4-hydroxyquinoline (HHQ), which is the precursor of PQS [20]. The *pqs* system controls the expression of *phzA-G* operons, which are closely associated with the production of pyocyanin, and regulates the production of extracellular DNA (eDNA), which is involved in the formation of stable and mature biofilms [21,22]. During the entire QS regulation process of *P. aeruginosa*, the LasR signal receptor is the earliest to be activated, regulating the expression of the most virulent or biofilm production-related genes, including other quorum-sensing receptors, which makes it a hot topic in this field [23].

Sulfur-containing organics are a class of compounds with a wide range of biological activities and functions that exist widely in oceans and on land. For instance, the marine-sourced natural products dithiolopyrrolone antibiotics and kottamide E have strong broad-spectrum antibacterial activity [24,25], isothiocyanates isolated from cruciferous plants can prevent cancers of the lungs and liver [26], brassilexin and methoxybrassitin were found to have antifungal activity, etc. [27,28,29] (Figure 1A). Researchers also found that garlic extracts could facilitate the rapid clearance of *P. aeruginosa* infection from the lungs of mice [30]. Further studies confirmed that the potent QS inhibitor in garlic is ajoene, which attenuates the main virulence factor regulated by QS [31]. Givskov et al. [32], by screening an internal compound library, found that disulfide-bonded derivatives structurally similar to ajoene showed QS inhibitory activity. Benzo heterocyclic skeleton is one of the most important structural components and biological compounds in pharmaceutical chemistry. Owing to the rich structural diversity of the bioactive benzo-heterocyclic ring, it has become an important structural component in many pharmaceutical preparations [33,34,35]. Both allicin and ajoene have sulfoxide structures, and we hypothesized that benzoheterocyclic sulfoxide might also have quorum sensing activity. In our previous work, we synthesized a series of benzoheterocyclic sulfoxide derivatives in the hope of finding new lead compounds to inhibit the quorum sensing of *P. aeruginosa* [36]. In addition, benzoheterocyclic sulfoxide derivatives in a planar π system in three-dimensional space structures were analyzed. We speculated that if the benzoheterocycle skeleton was stretched for phenyl heterocyclic sulfoxides, there might be multiple perspectives of three-dimensional rotation π systems. Owing to its longer skeleton structure, it may have a pocket advantage over the benzoheterocyclic structure from the perspective of protein docking (Figure 1B).

Hence, in this study, four series of phenyloxadiazole and phenyltetrazole sulfoxide derivatives were designed and synthesized to optimize potent *P. aeruginosa* QSIs. Through the biofilm inhibitory behavior of *P. aeruginosa* PAO1 and the fluorescence expression analysis of reporter strains (PAO1-*lasB*-*gfp*, PAO1-*rhlA*-*gfp*, and PAO1-*pqsA*-*gfp*), **5b** was found to be the most active biofilm inhibitor. We also tested **5b** for QS-activated virulence factors (elastase, rhamnolipid, and pyocyanin) production. Finally, we explored the binding effects between **5b** and the LasR receptor protein with molecular docking.

## 2. Results and Discussion

### 2.1. Chemistry

A series of phenyldioxazole and phenyltetrazole sulfoxide derivatives were synthesized. The preparation methods for the titled compounds are described in Figure 1. The different substituted benzyl bromides **2** were added dropwise to a solution of phenyldioxazole thiol or phenyltetrazole thiol **1** and Et_3_N in MeCN at room temperature to obtain intermediate thioethers **3**. Then, intermediates **3** were oxidized by *m*-chloroperoxybenzoic acid (*m*CPBA) in dichloromethane to obtain title compounds **4a**–**4o** (containing phenyltetrazole), **5a**–**5o** (containing phenyldioxazole), **6a**–**6o** (containing 4-methoxylphenyltetrazole), and **7a**–**7o** (containing 4-chlorophenyldioxazole) [37]. ^1^H NMR, ^13^C NMR spectra and HRMS spectra of all compounds are shown in the experimental section and Appendix A.

### 2.2. Evaluation of Inhibition of P. aeruginosa Biofilm and Structure–Activity Relationship (SAR) Studies

First, all target compounds were tested for their inhibitory activities against *P. aeruginosa* PAO1 biofilms (Table 1); 2-aminobenzimidazole was set as the positive control [38]. On the basis of previous studies, phenyldioxazole or phenyltetrazole sulfoxide derivatives were preferred for study. As shown in Table 1, in contrast to the good biofilm inhibitory activity in **5a**–**5o**, there was little or no effect observed in the **4a**–**4o** series compounds. We observed that the biofilm activity of most compounds in series **5** was higher than that of the positive control, which may be due to the introduction of five-membered rings by oxygen atoms. Among them, **5b** (R^2^ was substituted by *para*-chloro, inhibition rate was 46.85 ± 2.76%) and **5f** (R^2^ was substituted by 4-naphthyl; inhibition rate was 40.88 ± 0.75%) had the most significant inhibitory activity. However, the activities of **5c** and **5d** were weaker than those of **5b** (**5c**, 23.26 ± 3.20%; **5d**, 17.55 ± 2.85%), indicating that substitution of the *para*-position of the benzene ring may have better inhibitory activity. In addition, when R^2^ was substituted at the para-position of the benzene ring, the inhibitory activity of chlorinated derivative was higher than that of fluorinated derivative (**5h**, 23.17 ± 1.10%) and brominated derivative (**5i**, 21.94 ± 2.72%). When R^2^ were alkyl-branched chains, **5n** and **5o** showed no inhibitory activity (**5n**, 0.45 ± 0.19%; **5o**, 4.75 ± 0.96%), indicating that the aromatic ring was a necessary active group. Based on preliminary SAR analysis, only the *para*-substitution on the R^2^ benzene ring was conducive to activity; in particular, the *para*-chloro group promoted the activity. It should be noted that the three aryl groups were the optimal framework for designing phenyloxazole inhibitors.

Compared with the **4a**–**4o** series, the anti-biofilm activity was not significantly improved by introducing the R^1^ as *para*-methoxy group in **6a**–**6o** series derivatives. In the **7a**–**7o** series, the antibiofilm activity of replacing R^1^ with *para*-chloro substitution was not as good as that of the **5a**–**5o** series. In conclusion, when the *para*-chloro-substituted phenyloxazole ring and *para*-methoxyl-substituted phenyltetrazole were applied, the inhibitory activity against *P. aeruginosa* biofilm was not improved. These experimental results showed that the *para*-chloro substitution group was not essential for activity as R^1^ on the benzene ring compared to chloro as the R^2^ on the other side. In addition, we also tested the *P. aeruginosa* biofilm inhibitory activity of the corresponding intermediate thioethers of **5b** and **5f** (the biofilm inhibition rate of the intermediate thioether of **5b** was −3.77 ± 1.16%, and **5f** intermediate thioether was −5.04 ± 0.52%). These results revealed that the thioether derivatives of phenyloxazole had no PAO1 biofilm inhibitory activity, and they showed inhibitory activity against PAO1 biofilms only after being oxidized as sulfoxides. Therefore, the sulfoxide group was verified as an essential active functional group for inhibiting biofilm formation.

### 2.3. Effect of Sulfoxide Derivatives on QS System Reporter Strains

As previously mentioned, *las*, *rhl*, and *pqs* pathways are critical for the regulation of biofilm formation and the secretion of virulence factors in the QS system of *P. aeruginosa*. According to the regulatory process of the QS system in *P. aeruginosa*, Givskov et al. [18,39,40] reported the promoters in the QS pathway, including the *lasB* gene encoding elastase, which was shown to be under the transcriptional control of LasR; the first gene *rhlA* encodes the *rhl* operon of rhamnotransferase, and the first gene *pqsA* encodes the pqsABCDE of the PQS molecule. They were respectively fused with unstable green fluorescent protein (GFP) to construct three reporter strains, PAO1-*lasB*-*gfp*, PAO1-*rhlA*-*gfp*, and PAO1-*pqsA*-*gfp*, and a detection system for screening small molecule QSIs was established [41,42].

To further explore the mechanism of biofilm formation inhibition caused by the synthetic compounds, reporter strains PAO1-*lasB*-*gfp*, PAO1-*rhlA*-*gfp*, and PAO1-*pqsA*-*gfp* were introduced. Five compounds, **5a**, **5b**, **5f**, **5j**, and **5k**, which showed the best anti-biofilm activities, were selected for study (Figure 2). The experimental results verified that at the concentration of 20 μM, all five compounds could inhibit the fluorescence expression of the PAO1-*lasB*-*gfp* strain; of these compounds **5b** and **5f** had the best inhibitory effect (Figure 2A). Although **5a**, **5b**, **5f**, **5k**, and **5j** had certain inhibitory effects on the PAO1-*lasB*-*gfp* and PAO1-*pqsA*-*gfp* strains, the expression of *rhlA*-*gfp* was not particularly affected under the same experimental conditions (Figure 2B,C). The *las* system is upstream of the quorum sensing network [17], and biological reporter gene analysis indicated that phenyloxadiazole sulfoxide derivatives may specifically exhibited anti-biofilm activity via the *las* pathway.

### 2.4. The Effect of ***5b*** on PAO1-lasB-gfp and P. aeruginosa PAO1 Biofilm Growth and Formation

Based on the experimental results, **5b** was selected as the key research object. (Figure 3). Under the premise of no effect on the growth function of the PAO1-*lasB*-*gfp* reporter strain (Figure 3A), **5b** showed dose-dependent fluorescence inhibition of the PAO1-*lasB*-*gfp* strain at different concentrations of 20 μM, 10 μM, 5 μM, 2.5 μM, and 1.25 (Figure 3B). Based on the dose-response curves obtained, the IC_50_ value of **5b** against the PAO1-*lasB*-*gfp* strain was calculated as 3.53 ± 0.16 μM (Figure 3C). Similarly, we further verified the effect of **5b** on the growth and formation of the PAO1 biofilm. The effect of **5b** on the growth of *P. aeruginosa* PAO1 was assessed by monitoring the OD_600_ of the culture hourly. The results showed that the normal growth of *P. aeruginosa* PAO1 was not affected when the maximum concentration of **5b** was 50 μM (Figure 3D). Compound **5b** was incubated at concentrations of 50, 25, 12.5, 5, and 2.5 μM, and the control for 24 h and the OD values at 600 nm were evaluated before the biofilm experiments. We found that **5b** could reduce the biofilm formation in a dose-dependent manner (Figure 3E). In addition, the reduction of biofilm formation by **5b** was also observed by confocal laser scanning microscopy (CLSM) (Figure 3F). As shown, biofilm formed with 50 μM of **5b** was shallower than the control biofilm; the height of the biofilm formed with the control group, positive group, and **5b** were 35, 27, and 14 μm, respectively.

### 2.5. Effect of ***5b*** on Virulence Factors

Further, **5b** was used as a template to demonstrate that it inhibits biofilm formation through the *las* pathway of *P. aeruginosa*. We measured the effect of **5b** on the production of three virulence factors, elastase, pyocyanin, and rhamnolipid, in *P. aeruginosa* PAO1, which were regulated by *las*, *rhl*, and *pqs* systems, respectively. The results showed that **5b** could reduce elastase production in a concentration-dependent manner at a concentration of 50 μM, 25 μM, and 12.5 μM (Figure 4A). Compound **5b** inhibited pyocyanin production only at high concentrations and had little effect on rhamnolipid production. In summary, **5b** suppressed the expression of the QS system *lasB* gene and decreased the production of the virulence factor elastase. Thus, **5b** inhibited the formation of the PAO1 biofilm through the *las* pathway.

### 2.6. Molecular Docking Study

Furthermore, we explored the binding properties of **5b** and QS-associated proteins by molecular docking. As the autoinducer of the *las* pathway in *P. aeruginosa*, OdDHL binds to the cytoplasmic receptor protein LasR after synthesis by the LasI protein and forms the complex LasR-OdDHL, which binds to the promoter region of the target gene. The LasR-OdDHL complex then induces gene transcription of various virulence factors and specific proteins [43]. Silico molecular docking was performed to predict the binding models of **5b** and **5f** to the homologous signal receptor protein LasR (Figure 5, Table 2). The lowest binding energies of the docked conformations were selected from 30 hypothetical conformations as the modes for the corresponding compounds. The docking results are shown in Figure 5, where the benzene ring in the phenyloxazole structure of compound **5f** forms hydrophobic interactions with the residue Gly 126; the π bond in the naphthalene ring also developed hydrophobic interaction with the key residues Tyr 56, while the π bond in the chlorophenyl ring of **5b** had strong hydrophobic interactions with Trp 88 and Phe 101. It should be noted that the *para*-chlorinated benzene ring also formed binding interactions with the residue Leu 110, which demonstrates the importance of the *para*-chlorinated characteristics in phenyloxazole sulfoxide derivatives for *P. aeruginosa* biofilm activity. These results are consistent with the SAR results observed above and further interpret the active mechanism of phenyloxazole derivatives from the perspective of target and protein binding.

## 3. Materials and Methods

### 3.1. Chemistry

#### 3.1.1. General Methods for Synthetic Chemistry

All solvents and reagents were obtained from commercial sources without further purification. ^1^H and ^13^C NMR spectra were recorded on a Bruker Avance III 400 at 600 and 150 MHz or 400 and 100 MHz spectrometer. Chemical shifts were recorded as δ in units of parts per million (ppm), while tetramethylsilane (TMS) was used as an internal standard. Compounds were dissolved in CDCl_3_. Mass spectra were recorded on a SCIEX series X500B QTOF mass spectrometer. Thin-layer chromatography (TLC) was performed using Huanghai GF254 Silica gel plates. Column chromatography was performed using silica gel (200–300 mesh, Beijing, China) with a linear solvent gradient.

#### 3.1.2. General Synthesis Method for Compounds **4a**–**7o**

To a solution of heterocyclic thiol (2 mmol) and Et_3_N (3 mmol) dropwise in MeCN (10 mL) were added benzyl bromide (2.4 mmol). The mixture was stirred for 2–4 h at room temperature (monitored by TLC) and quenched with 6M HCl aqueous. The mixture was then extracted by EtOAc (10 mL × 3), saturated in aqueous brine (5 mL × 2), dried over Na_2_SO_4_, and concentrated to dryness. The crude residue was purified by silica gel column, followed by gradient elution with a petroleum ether/ethyl acetate mixture (50/1–30/1 ratio) to provide intermediate thioethers. Thioethers intermediate (1 mmol) and *m*-chloroperoxy-benzoic acid (1 mmol) were added to a solvent of CH_2_Cl_2_ (5 mL). The reaction mixture was stirred for 2–4 h at room temperature (monitored by TLC), and then washed with saturated aqueous NaHCO_3_ (5 mL × 2) and brine (5 mL × 2), dried over Na_2_SO_4_, and concentrated to dryness. The residue was directly loaded onto a silica gel column followed by gradient elution with petroleum ether/ethyl acetate mixture (30/1–10/1 ratio) to obtain target compounds (**4a**–**4o**, **5a**–**5o**, **6a**–**6o**, **7a**–**7o**). 

5-(benzylsulfinyl)-2-phenyl-2*H*-tetrazole (**4a**): white solid, yield 79%. ^1^H NMR (600 MHz, CDCl_3_) δ 7.56–7.51 (m, 1H, Ph-H), 7.46 (t, *J* = 7.8 Hz, 2H, Ph-H), 7.34–7.28 (m, 3H, Ph-H), 7.19–7.14 (m, 4H, Ph-H), 4.90–4.71 (m, 2H, CH_2_). ^13^C NMR (151 MHz, CDCl_3_) δ 156.12(N=C-N), 132.70(Ph-C), 131.05(Ph-C), 130.62(Ph-C, 2C), 129.60(Ph-C, 2C), 129.31(Ph-C), 129.18(Ph-C, 2C), 127.32(Ph-C), 125.02(Ph-C, 2C), 60.30(CH_2_). ESI-HRMS: calcd. for C_14_H_12_N_4_OS [M + H]^+^, 285.0805; found, 285.0803.

5-((4-chlorobenzyl) sulfinyl)-2-phenyl-2*H*-tetrazole (**4b**): white solid, yield 73%. ^1^H NMR (600 MHz, CDCl_3_) δ 7.56 (d, *J* = 7.4 Hz, 1H, Ph-H), 7.52 (d, *J* = 8.1 Hz, 2H, Ph-H), 7.33–7.29 (m, 2H, Ph-H), 7.29–7.25 (m, 2H, Ph-H), 7.15–7.10 (m, 2H, Ph-H), 4.84–4.69 (m, 2H, CH_2_). ^13^C NMR (151 MHz, CDCl_3_) δ 155.74(N=C-N), 135.65(Ph-C), 132.68(Ph-C), 131.90, (Ph-C, 2C) 131.14(Ph-C), 129.67(Ph-C, 2C), 129.35(Ph-C, 2C), 125.90(Ph-C), 124.91(Ph-C, 2C), 59.31(CH_2_). ESI-HRMS: calcd. for C_14_H_11_ClN_4_OS [M + H]^+^, 319.0415; found, 319.0419.

5-((3-chlorobenzyl) sulfinyl)-2-phenyl-2*H*-tetrazole (**4c**): yellow oily liquid, yield 79%. ^1^H NMR (600 MHz, CDCl_3_) δ 7.59–7.55 (m, 1H, Ph-H), 7.54–7.50 (m, 2H, Ph-H), 7.35–7.32 (m, 3H, Ph-H), 7.26–7.22 (m, 2H, Ph-H), 7.08 (dt, *J* = 7.5, 1.3 Hz, 1H, Ph-H), 4.85–4.67 (m, 2H, CH_2_). ^13^C NMR (151 MHz, CDCl_3_) δ 155.80(N=C-N), 135.02(Ph-C), 132.68(Ph-C), 131.19(Ph-C), 130.58(Ph-C), 130.39(Ph-C), 129.74(Ph-C, 2C), 129.59(Ph-C), 129.51(Ph-C), 128.77(Ph-C), 124.95(Ph-C, 2C), 59.37(CH_2_). ESI-HRMS: calcd. for C_14_H_11_ClN_4_OS [M + H]^+^, 319.0415; found, 319.0415.

5-((2-chlorobenzyl) sulfinyl)-2-phenyl-2*H*-tetrazole (**4d**): white solid, yield 83%. ^1^H NMR (600 MHz, CDCl_3_) δ 7.58–7.52 (m, 3H), 7.45–7.43 (m, 2H), 7.39 (dd, *J* = 8.5, 1.4 Hz, 1H), 7.31–7.29 (m, 2H), 7.22–7.19 (m, 1H), 5.14–4.89 (m, 2H, CH_2_). ^13^C NMR (151 MHz, CDCl_3_) δ 156.47(N=C-N), 135.21(Ph-C), 133.27(Ph-C), 132.75(Ph-C), 131.14(Ph-C), 130.91(Ph-C), 129.92(Ph-C), 129.90(Ph-C, 2C), 127.57(Ph-C), 125.96(Ph-C), 124.75(Ph-C, 2C), 58.17(CH_2_). ESI-HRMS: calcd. for C_14_H_11_ClN_4_OS [M + H]^+^, 319.0415; found, 319.0416.

5-((4-methylbenzyl) sulfinyl)-2-phenyl-2*H*-tetrazole (**4e**): white solid, yield 87%. ^1^H NMR (600 MHz, CDCl_3_) δ 7.53 (t, *J* = 7.5 Hz, 1H, Ph-H), 7.46 (t, *J* = 7.9 Hz, 2H, Ph-H), 7.22–7.16 (m, 2H, Ph-H), 7.09 (d, *J* = 8.0 Hz, 2H, Ph-H), 7.03 (d, *J* = 8.0 Hz, 2H, Ph-H), 4.87–4.67 (m, 2H, CH_2_), 2.32 (s, 3H, CH_3_). ^13^C NMR (151 MHz, CDCl_3_) δ 156.14(N=C-N), 139.33(Ph-C), 132.67(Ph-C), 130.90(Ph-C), 130.39(Ph-C, 2C), 129.77(Ph-C, 2C), 129.44(Ph-C, 2C), 124.89(Ph-C, 2C), 123.96(Ph-C), 60.01(CH_2_), 21.11(CH_3_). ESI-HRMS: calcd. for C_15_H_14_N_4_OS [M + H]^+^, 299.0961; found, 299.0964.

5-((naphthalen-2-ylmethyl) sulfinyl)-2-phenyl-2*H*-tetrazole (**4f**): white solid, yield 83%. ^1^H NMR (600 MHz, CDCl_3_) δ 7.81–7.78 (m, 1H, Ph-H), 7.75–7.71 (m, 2H, Ph-H), 7.67–7.64 (m, 1H, Ph-H), 7.53–7.48 (m, 2H, Ph-H), 7.44–7.41 (m, 1H, Ph-H), 7.29 (t, *J* = 8.0 Hz, 2H, Ph-H), 7.19 (dd, *J* = 8.4, 1.8 Hz, 1H, Ph-H), 7.06 (d, *J* = 7.5 Hz, 2H, Ph-H), 5.04–4.86 (m, 2H, CH_2_). ^13^C NMR (151 MHz, CDCl_3_) δ 156.16(N=C-N), 133.24(Ph-C), 133.13(Ph-C), 132.54(Ph-C), 130.87(Ph-C), 130.43(Ph-C), 129.38(Ph-C, 2C), 129.01(Ph-C), 127.93(Ph-C), 127.68(Ph-C), 127.18(Ph-C), 127.08(Ph-C), 126.83(Ph-C), 124.79(Ph-C, 2C), 124.47(Ph-C), 60.66(CH_2_). ESI-HRMS: calcd. for C_18_H_14_N_4_OS [M + Na]^+^, 357.0781; found, 357.0764.

5-((4-nitrobenzyl) sulfinyl)-2-phenyl-2*H*-tetrazole (**4g**): yellow solid, yield 70%. ^1^H NMR (600 MHz, CDCl_3_) δ 8.26–8.14 (m, 2H, Ph-H), 7.63–7.59 (m, 1H, Ph-H), 7.57–7.54 (m, 2H, Ph-H), 7.51–7.48 (m, 2H, Ph-H), 7.47–7.44 (m, 2H, Ph-H), 4.99–4.87 (m, 2H, CH_2_). ^13^C NMR (151 MHz, CDCl_3_) δ 155.32(N=C-N), 148.41(Ph-C), 134.94(Ph-C), 132.62(Ph-C, 2C), 131.72(Ph-C), 131.34(Ph-C), 129.84(Ph-C, 2C), 124.85(Ph-C, 2C), 124.07(Ph-C, 2C), 58.72(CH_2_). ESI-HRMS: calcd. for C_14_H_11_N_5_O_3_S [M + H]^+^, 330.0655; found, 330.0653.

5-((4-fluorobenzyl) sulfinyl)-2-phenyl-2*H*-tetrazole (**4h**): white solid, yield 76%. ^1^H NMR (600 MHz, CDCl_3_) δ 7.57–7.49 (m, 3H, Ph-H), 7.33–7.30 (m, 2H, Ph-H), 7.21–7.17 (m, 2H, Ph-H), 7.01–6.98 (m, 2H, Ph-H), 4.84–4.70 (m, 2H CH_2_). ^13^C NMR (151 MHz, CDCl_3_) δ 164.12(Ph-C), 162.47(Ph-C), 155.82(N=C-N), 132.72(Ph-C), 132.52(Ph-C), 132.46(Ph-C), 131.14(Ph-C), 129.68(Ph-C, 2C), 124.93(Ph-C, 2C), 116.32(Ph-C), 116.17(Ph-C), 59.15(CH_2_). ESI-HRMS: calcd. for C_14_H_11_FN_4_OS [M + Na]^+^, 325.0530; found, 325.0527.

5-((4-bromobenzyl) sulfinyl)-2-phenyl-2*H*-tetrazole (**4i**): white solid, yield 88%. ^1^H NMR (600 MHz, CDCl_3_) δ 7.58–7.55 (m, 1H, Ph-H), 7.53–7.49 (m, 2H, Ph-H), 7.44–7.40 (m, 2H, Ph-H), 7.34–7.28 (m, 2H, Ph-H), 7.07–7.04 (m, 2H, Ph-H), 4.82–4.66 (m, 2H, CH_2_). ^13^C NMR (151 MHz, CDCl_3_) δ 155.73(N=C-N), 132.65(Ph-C), 132.29(Ph-C, 2C), 132.14(Ph-C, 2C), 131.11(Ph-C), 129.64(Ph-C, 2C), 126.40(Ph-C), 124.89(Ph-C, 2C), 123.81(Ph-C), 59.39(CH_2_). ESI-HRMS: calcd. for C_14_H_11_BrN_4_OS [M + Na]^+^, 384.9729; found, 384.9728.

2-phenyl-5-((4-(trifluoromethyl) benzyl) sulfinyl)-2*H*-tetrazole (**4j**): white solid, yield 70%. ^1^H NMR (600 MHz, CDCl_3_) δ 7.58–7.55 (m, 3H, Ph-H), 7.52–7.47 (m, 2H, Ph-H), 7.34 (d, *J* = 8.1 Hz, 2H, Ph-H), 7.31–7.28 (m, 2H, Ph-H), 4.92–4.77 (m, 2H, CH_2_). ^13^C NMR (151 MHz, CDCl_3_) δ 155.55(N=C-N), 132.62(Ph-C), 131.62(Ph-C), 131.59(Ph-C), 131.38(Ph-C, 2C), 131.20(Ph-C), 131.06(Ph-C), 129.69(Ph-C, 2C), 126.00(Ph-C), 125.97(Ph-C), 124.88(Ph-C, 2C), 59.41(CH_2_). ESI-HRMS: calcd. for C_15_H_11_F_3_N_4_OS [M + Na]^+^, 375.0498; found, 375.0492.

5-((3-methoxybenzyl) sulfinyl)-2-phenyl-2*H*-tetrazole (**4k**): yellow solid, yield 77%. ^1^H NMR (600 MHz, CDCl_3_) δ 7.58–7.52 (m, 1H, Ph-H), 7.50–7.43 (m, 2H, Ph-H), 7.25–7.17 (m, 3H, Ph-H), 6.88–6.86 (m, 1H, Ph-H), 6.74 (dt, *J* = 7.5, 1.2 Hz, 1H, Ph-H), 6.68–6.61 (m, 1H, Ph-H), 4.87–4.68 (m, 2H, CH_2_), 3.71 (s, 3H, CH_3_). ^13^C NMR (151 MHz, CDCl_3_) δ 160.05(Ph-C), 156.22(N=C-N), 132.71(Ph-C), 131.03(Ph-C), 130.19(Ph-C), 129.60(Ph-C, 2C), 128.59(Ph-C), 125.00(Ph-C, 2C), 122.71(Ph-C), 115.46(Ph-C), 115.39(Ph-C), 60.44(CH_2_), 55.30(OCH_3_). ESI-HRMS: calcd. for C_15_H_14_N_4_O_2_S [M + H]^+^, 315.0910; found, 315.0907.

5-((4-methoxybenzyl) sulfinyl)-2-phenyl-2*H*-tetrazole (**4l**): yellow oily liquid, yield 85%. ^1^H NMR (600 MHz, CDCl_3_) δ 7.54 (t, *J* = 7.5 Hz, 1H, Ph-H), 7.49–7.46 (m, 2H, Ph-H), 7.24–7.22 (m, 2H, Ph-H), 7.09–7.06 (m, 2H, Ph-H), 6.81–6.79 (m, 2H, Ph-H), 4.85–4.65 (m, 2H, CH_2_), 3.77 (s, 3H, CH_3_). ^13^C NMR (151 MHz, CDCl_3_) δ 160.40(Ph-C), 156.22(N=C-N), 132.76(Ph-C), 131.87(Ph-C, 2C), 130.98(Ph-C), 129.56(Ph-C, 2C), 124.95(Ph-C, 2C), 118.86(Ph-C), 114.58(Ph-C, 2C), 59.74(CH_2_), 55.33(OCH_3_). ESI-HRMS: calcd. for C_15_H_14_N_4_O_2_S [M + Na]^+^, 337.0730; found, 337.0722.

5-((3, 5-dimethoxybenzyl) sulfinyl)-2-phenyl-2*H*-tetrazole (**4m**): yellow solid, yield 56%. ^1^H NMR (600 MHz, CDCl_3_) δ 7.55–7.46 (m, 4H), 7.25 (d, *J* = 1.4 Hz, 1H), 6.39 (s, 1H, Ph-H), 6.26 (d, *J* = 2.2 Hz, 2H, Ph-H), 4.82–4.64 (m, 2H, CH_2_), 3.69 (s, 6H, CH_3_). ^13^C NMR (151 MHz, CDCl_3_) δ 161.21(Ph-C, 2C), 156.34(N=C-N), 132.71(Ph-C), 131.01(Ph-C), 129.60(Ph-C, 2C), 129.15(Ph-C, 2C), 124.97(Ph-C), 108.08(Ph-C, 2C), 101.60(Ph-C), 60.72(CH_2_), 55.43(OCH_3_, 2C). ESI-HRMS: calcd. for C_16_H_16_N_4_O_3_S [M + H]^+^, 345.1016; found, 345.1016.

Methyl 2-((2-phenyl-2*H*-tetrazol-5-yl) sulfinyl) acetate (**4n**): white solid, yield 55%. ^1^H NMR (600 MHz, CDCl_3_) δ 7.76–7.70 (m, 2H, Ph-H), 7.66–7.63 (m, 3H, Ph-H), 4.93–4.50 (m, 2H, CH_2_), 3.76 (s, 3H, CH_3_). ^13^C NMR (151 MHz, CDCl_3_) δ 164.91(Ph-C), 156.30(N=C-N), 132.77(Ph-C), 131.41(Ph-C), 130.10(Ph-C, 2C), 125.00(Ph-C, 2C), 56.32(CH_2_), 53.31(CH_3_). ESI-HRMS: calcd. for C_10_H_10_N_4_O_3_S [M + Na]^+^, 289.0366; found, 289.0364.

Ethyl 2-((2-phenyl-2*H*-tetrazol-5-yl) sulfinyl) (**4o**): acetate yellow oily liquid, yield 74% ^1^H NMR (600 MHz, CDCl_3_) δ 7.72 (dd, J = 7.4, 2.0 Hz, 2H, Ph-H), 7.65–7.63 (m, 3H, Ph-H), 4.93–4.48 (m, 2H, CH_2_), 4.20 (tt, *J* = 7.1, 3.5 Hz, 2H, CH_2_), 1.26 (t, *J* = 7.1 Hz, 3H, CH_3_). ^13^C NMR (151 MHz, CDCl_3_) δ 164.43(Ph-C), 156.36(N=C-N), 132.78(Ph-C), 131.38(Ph-C), 130.09(Ph-C, 2C), 124.98(Ph-C, 2C), 62.78(CH_2_), 56.49(CH_2_), 13.95(CH_3_). ESI-HRMS: calcd. for C_11_H_12_N_4_O_3_S [M + Na]^+^, 303.0522; found, 303.0520.

2-(benzylsulfinyl)-5-phenyl-1,3,4-oxadiazole (**5a**): white solid, yield 77%. ^1^H NMR (600 MHz, CDCl_3_) δ 8.08–8.00 (m, 2H), 7.60 (t, *J* = 7.5 Hz, 1H, Ph-H), 7.53 (dd, *J* = 8.4, 7.0 Hz, 2H, Ph-H), 7.37–7.33 (m, 3H, Ph-H), 7.31 (dt, *J* = 5.3, 3.6 Hz, 2H, Ph-H), 4.73–4.56 (m, 2H, CH_2_). ^13^C NMR (151 MHz, CDCl_3_) δ 167.16(C=N), 165.42(C=N), 132.82(Ph-C), 130.35(Ph-C, 2C), 129.24(Ph-C, 2C), 129.16(Ph-C, 2C), 127.84(Ph-C), 127.46(Ph-C, 2C), 122.58(Ph-C), 60.51(CH_2_). ESI-HRMS: calcd. for C_15_H_12_N_2_O_2_S [M + H]^+^, 285.0692; found, 285.0691.

2-((4-chlorobenzyl) sulfinyl)-5-phenyl-1,3,4-oxadiazole (**5b**): white solid, yield 81%. ^1^H NMR (600 MHz, CDCl_3_) δ 8.05–8.03 (m, 2H, Ph-H), 7.57 (dt, *J* = 41.7, 7.4 Hz, 3H, Ph-H), 7.35–7.31 (m, 2H, Ph-H), 7.26 (d, *J* = 8.5 Hz, 2H, Ph-H), 4.72–4.50 (m, 2H, CH_2_). ^13^C NMR (151 MHz, CDCl_3_) δ 167.31(C=N), 165.16(C=N), 135.61(Ph-C), 132.91(Ph-C), 131.74(Ph-C, 2C), 129.37(Ph-C, 2C), 129.30(Ph-C, 2C), 127.45(Ph-C, 2C), 126.37(Ph-C), 122.47(Ph-C), 59.49(CH_2_). ESI-HRMS: calcd. for C_15_H_11_ClN_2_O_2_S [M + Na]^+^, 341.0122; found, 341.0099.

2-((3-chlorobenzyl) sulfinyl)-5-phenyl-1,3,4-oxadiazole (**5c**): white solid, yield 79%. ^1^H NMR (600 MHz, CDCl_3_) δ 8.07–8.04 (m, 2H, Ph-H), 7.62–7.58 (m, 1H, Ph-H), 7.53 (dd, *J* = 8.4, 7.0 Hz, 2H, Ph-H), 7.36–7.32 (m, 2H, Ph-H), 7.28 (t, *J* = 7.9 Hz, 1H, Ph-H), 7.21 (dt, *J* = 7.6, 1.4 Hz, 1H, Ph-H), 4.77–4.47 (m, 2H, CH_2_). ^13^C NMR (151 MHz, CDCl_3_) δ 167.27(C=N), 165.21(C=N), 134.96(Ph-C), 132.89(Ph-C), 130.44(Ph-C), 130.33(Ph-C), 129.92(Ph-C), 129.44(Ph-C), 129.26(Ph-C, 2C), 128.59(Ph-C), 127.44(Ph-C, 2C), 122.44(Ph-C), 59.59(CH_2_). ESI-HRMS: calcd. for C_15_H_11_ClN_2_O_2_S [M + Na]^+^, 341.0122; found, 319.0102.

2-((2-chlorobenzyl) sulfinyl)-5-phenyl-1,3,4-oxadiazole (**5d**): white solid, yield 72%. ^1^H NMR (600 MHz, CDCl_3_) δ 8.10–8.06 (m, 2H, Ph-H), 7.64–7.59 (m, 1H, Ph-H), 7.56–7.52 (m, 2H, Ph-H), 7.44–7.42 (m, 2H), 7.33 (td, *J* = 7.7, 1.7 Hz, 1H, Ph-H), 7.28 (dd, *J* = 7.6, 1.4 Hz, 1H, Ph-H), 5.00–4.77 (m, 2H, CH_2_). ^13^C NMR (151 MHz, CDCl_3_) δ 167.23(C=N), 165.40(C=N), 135.03(Ph-C), 132.96(Ph-C), 132.87(Ph-C), 130.81(Ph-C), 130.02(Ph-C), 129.26(Ph-C, 2C), 127.49(Ph-C), 127.47(Ph-C,2C), 126.04(Ph-C), 122.56(Ph-C), 58.12(CH_2_). ESI-HRMS: calcd. for C_15_H_11_ClN_2_O_2_S [M + Na]^+^, 341.0122; found, 319.0100.

2-((4-methylbenzyl) sulfinyl)-5-phenyl-1,3,4-oxadiazole (**5e**): white solid, yield 80%. ^1^H NMR (600 MHz, CDCl_3_) δ 8.07–8.03 (m, 2H, Ph-H), 7.62–7.58 (m, 1H, Ph-H), 7.53 (dd, *J* = 8.4, 6.9 Hz, 2H, Ph-H), 7.18 (d, *J* = 8.1 Hz, 2H, Ph-H), 7.14 (d, *J* = 7.9 Hz, 2H, Ph-H), 4.76–4.57 (m, 2H, CH_2_), 2.30 (s, 3H, CH_3_). ^13^C NMR (151 MHz, CDCl_3_) δ 167.10(C=N), 165.47(C=N), 139.31(Ph-C), 132.78(Ph-C), 130.21(Ph-C, 2C), 129.88(Ph-C, 2C), 129.23(Ph-C, 2C), 127.45(Ph-C, 2C), 124.63(Ph-C), 122.63(Ph-C), 60.31(CH_2_), 21.18(CH_3_). ESI-HRMS: calcd. for C_16_H_14_N_2_O_2_S [M + H]^+^, 299.0849; found, 299.0849.

2-((naphthalen-2-ylmethyl) sulfinyl)-5-phenyl-1,3,4-oxadiazole (**5f**): yellow solid, yield 67%. ^1^H NMR (600 MHz, CDCl_3_) δ 8.00–7.91 (m, 2H, Ph-H), 7.85–7.72 (m, 4H, Ph-H), 7.56 (t, *J* = 7.5 Hz, 1H, Ph-H), 7.52–7.44 (m, 4H, Ph-H), 7.36 (dd, *J* = 8.4, 1.7 Hz, 1H, Ph-H), 4.90–4.74 (m, 2H, CH_2_). ^13^C NMR (151 MHz, CDCl_3_) δ 167.21(C=N), 165.48(C=N), 133.35(Ph-C), 133.26(Ph-C), 132.76(Ph-C), 130.28(Ph-C), 129.19(Ph-C, 2C), 129.06(Ph-C), 128.02(Ph-C), 127.73(Ph-C), 127.42(Ph-C, 2C), 127.12(Ph-C), 126.91(Ph-C), 126.71(Ph-C), 125.21(Ph-C), 122.51(Ph-C), 60.83(CH_2_). ESI-HRMS: calcd. for C_19_H_14_N_2_O_2_S [M + Na]^+^, 357.0668; found, 357.0641.

2-((4-nitrobenzyl) sulfinyl)-5-phenyl-1,3,4-oxadiazole (**5g**): yellow solid, yield 63%. ^1^H NMR (600 MHz, CDCl_3_) δ 8.22 (d, *J* = 8.6 Hz, 2H, Ph-H), 8.05 (s, 1H, Ph-H), 7.61 (d, *J* = 7.4 Hz, 1H, Ph-H), 7.54 (t, *J* = 8.5 Hz, 5H, Ph-H), 4.80–4.70 (m, 2H, CH_2_). ^13^C NMR (151 MHz, CDCl_3_) δ 167.55(C=N), 164.76(C=N), 148.45(Ph-C), 135.11(Ph-C), 133.09(Ph-C), 131.60(Ph-C, 2C), 129.36(Ph-C, 2C), 127.43(Ph-C, 2C), 124.10(Ph-C, 2C), 122.29(Ph-C), 59.00(CH_2_). ESI-HRMS: calcd. for C_15_H_11_N_3_O_4_S [M + Na]^+^, 352.0362; found, 352.0350.

2-((4-fluorobenzyl) sulfinyl)-5-phenyl-1,3,4-oxadiazole (**5h**): white solid, yield 72%. ^1^H NMR (600 MHz, CDCl_3_) δ 8.11–8.03 (m, 2H, Ph-H), 7.67–7.50 (m, 3H, Ph-H), 7.35–7.28 (m, 2H, Ph-H), 7.05 (t, *J* = 8.6 Hz, 2H, Ph-H), 4.71–4.55 (m, 2H, CH_2_). ^13^C NMR (151 MHz, CDCl_3_) δ 167.28(C=N), 165.25(C=N), 162.51(Ph-C), 132.90(Ph-C), 132.31(Ph-C), 132.25(Ph-C), 129.29(Ph-C, 2C), 127.44(Ph-C, 2C), 123.71(Ph-C), 122.50(Ph-C), 116.33(Ph-C), 116.19(Ph-C), 59.42(CH_2_). ESI-HRMS: calcd. for C_15_H_11_FN_2_O_2_S [M + Na]^+^, 325.0417; found, 325.0410.

2-((4-bromobenzyl) sulfinyl)-5-phenyl-1,3,4-oxadiazole (**5i**): white solid, yield 84%. ^1^H NMR (600 MHz, CDCl_3_) δ 8.06–7.94 (m, 2H, Ph-H), 7.64–7.58 (m, 1H, Ph-H), 7.53 (dd, *J* = 8.4, 7.0 Hz, 2H, Ph-H), 7.50–7.45 (m, 2H, Ph-H), 7.22–7.15 (m, 2H, Ph-H), 4.74–4.43 (m, 2H, CH_2_). ^13^C NMR (151 MHz, CDCl_3_) δ 167.27(C=N), 165.12(C=N), 132.89(Ph-C), 132.31(Ph-C, 2C), 131.99(Ph-C, 2C), 129.29(Ph-C, 2C), 127.43(Ph-C, 2C), 126.87(Ph-C), 123.79(Ph-C), 122.42(Ph-C), 59.51(CH_2_). ESI-HRMS: calcd. for C_15_H_11_BrN_2_O_2_S [M + Na]^+^, 384.9617; found, 384.9594.

2-phenyl-5-((4-(trifluoromethyl) benzyl) sulfinyl)-1,3,4-oxadiazole (**5j**): white solid, yield 70%. ^1^H NMR (600 MHz, CDCl_3_) δ 8.06–8.02 (m, 2H, Ph-H), 7.66–7.61 (m, 3H, Ph-H), 7.55 (dd, *J* = 8.5, 7.0 Hz, 2H, Ph-H), 7.49 (d, *J* = 8.0 Hz, 2H, Ph-H), 4.83–4.62 (m, 2H, CH_3_). ^13^C NMR (151 MHz, CDCl_3_) δ 167.41(C=N), 165.01(C=N), 132.98(Ph-C), 132.02(Ph-C), 131.34(Ph-C, 2C), 130.91(Ph-C), 129.30(Ph-C, 2C), 127.44(Ph-C, 2C), 126.01(Ph-C), 124.64(Ph-C), 122.84(Ph-C), 122.38(Ph-C), 59.54(CH_2_). ESI-HRMS: calcd. for C_16_H_11_F_3_N_2_O_2_S [M + H]^+^, 353.0566; found, 353.0566.

2-((3-methoxybenzyl) sulfinyl)-5-phenyl-1,3,4-oxadiazole (**5k**): white solid, yield 74%. ^1^H NMR (600 MHz, CDCl_3_) δ 8.16–8.05 (m, 2H, Ph-H), 7.68–7.52 (m, 3H, Ph-H), 7.29–7.27 (m, 1H, Ph-H), 6.94–6.82 (m, 3H, Ph-H), 4.75–4.53 (m, 2H, CH_2_), 3.77 (s, 3H, CH_3_). ^13^C NMR (151 MHz, CDCl_3_) δ 167.16(C=N), 165.50(C=N), 160.03(Ph-C), 132.83(Ph-C), 130.21(Ph-C), 129.25(Ph-C, 2C), 129.18(Ph-C), 127.46(Ph-C, 2C), 122.59(Ph-C), 122.52(Ph-C), 115.60(Ph-C), 115.08(Ph-C), 60.65(CH_2_), 55.26(OCH_3_). ESI-HRMS: calcd. for C_16_H_14_N_2_O_3_S [M + H]^+^, 315.0798; found, 315.0782.

2-((4-methoxybenzyl) sulfinyl)-5-phenyl-1,3,4-oxadiazole (**5l**): white solid, yield 57%. ^1^H NMR (600 MHz, CDCl_3_) δ 8.10–8.02 (m, 2H, Ph-H), 7.61–7.48 (m, 3H, Ph-H), 7.21 (d, *J* = 8.5 Hz, 2H, Ph-H), 6.89–6.81 (m, 2H, Ph-H), 4.67–4.55 (m, 2H, CH_2_), 3.75 (s, 3H, CH_3_). ^13^C NMR (151 MHz, CDCl_3_) δ 167.12(C=N), 165.50(C=N), 160.39(Ph-C), 132.79(Ph-C), 131.63(Ph-C, 2C), 129.24(Ph-C, 2C), 127.45(Ph-C, 2C), 122.63(Ph-C), 119.49(Ph-C), 114.64(Ph-C, 2C), 60.03(CH_2_), 55.27(OCH_3_). ESI-HRMS: calcd. for C_16_H_14_N_2_O_3_S [M + Na]^+^, 337.0617; found, 337.0601.

2-((3,5-dimethoxybenzyl) sulfinyl)-5-phenyl-1,3,4-oxadiazole (**5m**): yellow solid, yield 69%. ^1^H NMR (600 MHz, CDCl_3_) δ 8.07 (d, *J* = 7.8 Hz, 2H, Ph-H), 7.60 (t, *J* = 7.4 Hz, 1H, Ph-H), 7.53 (dd, *J* = 8.3, 7.0 Hz, 2H, Ph-H), 6.42 (dd, *J* = 10.3, 2.3 Hz, 3H, Ph-H), 4.65–4.54 (m, 2H, CH_2_), 3.71 (d, J = 0.8 Hz, 6H, CH_3_). ^13^C NMR (151 MHz, CDCl_3_) δ 167.15(C=N), 165.51(C=N), 161.19(Ph-C), 132.84(Ph-C), 129.77(Ph-C, 2C), 129.25(Ph-C, 2C), 127.43(Ph-C), 125.69(Ph-C), 122.53(Ph-C), 108.11(Ph-C, 2C), 101.35(Ph-C), 60.87(CH_2_), 55.35(OCH_3_, 2C). ESI-HRMS: calcd. for C_17_H_16_N_2_O_4_S [M + H]^+^, 345.0904; found, 345.0894.

Methyl 2-((5-phenyl-1,3,4-oxadiazol-2-yl) sulfinyl) acetate (**5n**): white solid, yield 75%. ^1^H NMR (600 MHz, CDCl_3_) δ 8.18–8.11 (m, 2H, Ph-H), 7.69–7.60 (m, 1H, Ph-H), 7.58 (dd, *J* = 8.4, 6.9 Hz, 2H, Ph-H), 4.63–4.32 (m, 2H, CH_2_), 3.85 (s, 3H, CH_3_). ^13^C NMR (151 MHz, CDCl_3_) δ 167.47(C=N), 165.37(C=N), 164.36(C=O), 132.99(Ph-C), 129.32(Ph-C, 2C), 127.53(Ph-C, 2C), 122.48(Ph-C), 57.09(CH_2_), 53.42(CH_3_). ESI-HRMS: calcd. For C_11_H_10_N_2_O_4_S [M + Na]^+^, 289.0253; found, 289.0239.

Ethyl 2-((5-phenyl-1,3,4-oxadiazol-2-yl) sulfinyl) acetate (**5o**): white solid, yield 61%. ^1^H NMR (600 MHz, CDCl_3_) δ 8.20–8.10 (m, 2H, Ph-H), 7.68–7.61 (m, 1H, Ph-H), 7.58 (d, *J* = 7.9 Hz, 2H, Ph-H), 4.58–4.39 (m, 2H, CH_2_), 4.30 (q, *J* = 7.1 Hz, 2H, CH_2_), 1.32–1.28 (m, 3H, CH_3_). ^13^C NMR (151 MHz, CDCl_3_) δ 168.03(C=N), 166.01(C=N), 164.32(C=O), 133.54(Ph-C), 129.88(Ph-C, 2C), 128.11(Ph-C, 2C), 123.08(Ph-C), 63.46(CH_2_), 57.92(CH_2_), 14.55(CH_3_). ESI-HRMS: calcd. for C_12_H_12_N_2_O_4_S [M + Na]^+^, 303.0410; found, 303.0394.

5-(benzylsulfinyl)-2-(4-methoxyphenyl)-2*H*-tetrazole (**6a**): white solid, yield 84%. ^1^H NMR (600 MHz, CDCl_3_) δ 7.38–7.29 (m, 3H, Ph-H), 7.17 (d, *J* = 7.5 Hz, 2H, Ph-H), 7.07 (d, *J* = 8.9 Hz, 2H, Ph-H), 6.93 (d, *J* = 8.9 Hz, 2H, Ph-H), 4.89–4.70 (m, 2H, CH_2_), 3.85 (s, 3H, CH_3_). ^13^C NMR (151 MHz, CDCl_3_) δ 161.41(Ph-C), 156.08(N=C-N), 130.62(Ph-C, 2C), 129.26(Ph-C), 129.15(Ph-C, 2C), 127.41(Ph-C), 126.51(Ph-C, 2C), 125.33(Ph-C), 114.65(Ph-C, 2C), 60.17(CH_2_), 55.69(OCH_3_). ESI-HRMS: calcd. for C_15_H_14_N_4_O_2_S [M + H]^+^, 315.0910; found, 315.0902. 

5-((4-chlorobenzyl) sulfinyl)-2-(4-methoxyphenyl)-2*H*-tetrazole (**6b**): white solid, yield 87%. ^1^H NMR (600 MHz, CDCl_3_) δ 7.27 (d, *J* = 7.8 Hz, 2H, Ph-H), 7.21 (dd, *J* = 8.7, 1.3 Hz, 2H, Ph-H), 7.16–7.11 (m, 2H, Ph-H), 6.97 (dd, *J* = 8.7, 1.3 Hz, 2H, Ph-H), 4.83–4.67 (m, 2H, CH_2_), 3.87 (d, *J* = 1.3 Hz, 3H, CH_3_). ^13^C NMR (151 MHz, CDCl_3_) δ 161.47(Ph-C), 155.69(N=C-N), 135.58(Ph-C), 131.91(Ph-C, 2C), 129.32(Ph-C, 2C), 126.41(Ph-C, 2C), 126.02(Ph-C), 125.31(Ph-C), 114.70(Ph-C, 2C), 59.17(CH_2_), 55.72(OCH_3_). ESI-HRMS: calcd. for C_15_H_13_ClN_4_O_2_S [M + Na]^+^, 371.0340; found, 371.0318.

2-(4-methoxyphenyl)-5-((4-methylbenzyl) sulfinyl)-2*H*-tetrazole (**6e**): white solid, yield 81%. ^1^H NMR (600 MHz, CDCl_3_) δ 7.09 (dd, *J* = 8.4, 5.8 Hz, 4H, Ph-H), 7.03 (d, *J* = 7.9 Hz, 2H, Ph-H), 6.95–6.91 (m, 2H, Ph-H), 4.85–4.66 (m, 2H, CH_2_), 3.85 (s, 3H, CH_3_), 2.32 (s, 3H, CH_3_). ^13^C NMR (151 MHz, CDCl_3_) δ 161.38(Ph-C), 156.20(N=C-N), 139.34(Ph-C), 130.48(Ph-C, 2C), 129.82(Ph-C, 2C), 126.46(Ph-C, 2C), 125.39(Ph-C), 124.15(Ph-C), 114.59(Ph-C, 2C), 59.94(CH_2_), 55.69(OCH_3_), 21.18(CH_3_). ESI-HRMS: calcd. for C_16_H_16_N_4_O_2_S [M + Na]^+^, 351.0886; found, 351.0860.

2-(4-methoxyphenyl)-5-((naphthalen-2-ylmethyl) sulfinyl)-2*H*-tetrazole (**6f**): yellow solid, yield 83%. ^1^H NMR (600 MHz, CDCl_3_) δ 7.81–7.78 (m, 1H, Ph-H), 7.73 (dd, *J* = 8.6, 5.6 Hz, 2H, Ph-H), 7.65 (d, *J* = 1.7 Hz, 1H, Ph-H), 7.50 (td, *J* = 7.3, 1.6 Hz, 2H, Ph-H), 7.18 (dd, *J* = 8.5, 1.7 Hz, 1H, Ph-H), 7.01–6.89 (m, 2H, Ph-H), 6.81–6.70 (m, 2H, Ph-H), 5.02–4.84 (m, 2H, CH_2_), 3.78 (s, 3H, CH_3_). ^13^C NMR (151 MHz, CDCl_3_) δ 161.21(Ph-C), 156.17(N=C-N), 133.24(Ph-C), 133.15(Ph-C), 130.44(Ph-C), 129.00(Ph-C), 127.96(Ph-C), 127.70(Ph-C), 127.22(Ph-C), 127.06(Ph-C), 126.81(Ph-C), 126.27(Ph-C, 2C), 125.21(Ph-C), 124.60(Ph-C), 114.43(Ph-C, 2C), 60.56(CH_2_), 55.62(OCH_3_). ESI-HRMS: calcd. for C_19_H_16_N_4_O_2_S [M + Na]^+^, 387.0886; found, 387.0866.

2-(4-methoxyphenyl)-5-((4-nitrobenzyl) sulfinyl)-2*H* tetrazole (**6g**): yellow solid, yield 85%. ^1^H NMR (600 MHz, CDCl_3_) δ 8.20–8.15 (m, 2H, Ph-H), 7.50–7.45 (m, 2H, Ph-H), 7.33 (d, *J* = 9.0 Hz, 2H, Ph-H), 6.99 (d, *J* = 9.0 Hz, 2H, Ph-H), 4.95–4.86 (m, 2H, CH_2_), 3.87 (s, 3H, CH_3_). ^13^C NMR (151 MHz, CDCl_3_) δ 161.62(Ph-C), 155.28(N=C-N), 148.40(Ph-C), 135.05(Ph-C), 131.75(Ph-C, 2C), 126.37(Ph-C, 2C), 125.21(Ph-C), 124.04(Ph-C, 2C), 114.86(Ph-C, 2C), 58.63(CH_2_), 55.75(OCH_3_). ESI-HRMS: calcd. for C_15_H_13_N_5_O_4_S [M + Na]^+^, 382.0580; found, 382.0566.

5-((4-fluorobenzyl) sulfinyl)-2-(4-methoxyphenyl)-2*H*-tetrazole (**6h**): white solid, yield 80%. ^1^H NMR (600 MHz, CDCl_3_) δ 7.24–7.20 (m, 4H, Ph-H), 7.03–6.98 (m, 4H, Ph-H), 4.88–4.73 (m, 2H, CH_2_), 3.88 (s, 3H, CH_3_). ^13^C NMR (151 MHz, CDCl_3_) δ 164.09(Ph-C), 162.44(Ph-C), 161.48(Ph-C), 155.76(N=C-N), 132.47, (Ph-C) 126.43(Ph-C, 2C), 125.34(Ph-C), 123.39(Ph-C), 116.28(Ph-C), 116.14(Ph-C), 114.73(Ph-C, 2C), 59.01(CH_2_), 55.71(OCH_3_). ESI-HRMS: calcd. for C_15_H_13_FN_4_O_2_S [M + Na]^+^, 355.0635; found, 355.0611.

5-((4-bromobenzyl) sulfinyl)-2-(4-methoxyphenyl)-2*H*-tetrazole (**6i**): white solid, yield 89%. ^1^H NMR (600 MHz, CDCl_3_) δ 7.46–7.39 (m, 2H, Ph-H), 7.24–7.18 (m, 2H, Ph-H), 7.09–7.04 (m, 2H, Ph-H), 7.00–6.95 (m, 2H, Ph-H), 4.81–4.64 (m, 2H, CH_2_), 3.87 (s, 3H, CH_3_). ^13^C NMR (151 MHz, CDCl_3_) δ 161.48(Ph-C), 155.66(N=C-N), 132.30(Ph-C, 2C), 132.17(Ph-C, 2C), 126.51(Ph-C, 2C), 126.42(Ph-C), 125.31(Ph-C), 123.79(Ph-C), 114.70(Ph-C, 2C), 59.29(CH_2_), 55.72(OCH_3_). ESI-HRMS: calcd. for C_15_H_13_BrN_4_O_2_S [M + Na]^+^, 414.9835; found 414.9834.

2-(4-methoxyphenyl)-5-((4-(trifluoromethyl) benzyl) sulfinyl) -2*H*-tetrazole (**6g**): yellow solid, yield 83%. ^1^H NMR (600 MHz, CDCl_3_) δ 7.59 (d, *J* = 8.0 Hz, 2H, Ph-H), 7.36 (d, *J* = 8.0 Hz, 2H, Ph-H), 7.24–7.19 (m, 2H, Ph-H), 7.00–6.94 (m, 2H, Ph-H), 5.01–4.67 (m, 2H, CH_2_), 3.87 (s, 3H, CH_3_). ^13^C NMR (151 MHz, CDCl_3_) δ 161.51(Ph-C), 155.57(N=C-N), 131.78(Ph-C, 2C), 131.52(Ph-C), 131.31(Ph-C), 131.08(Ph-C), 126.38(Ph-C, 2C), 125.98(Ph-C), 125.25(Ph-C), 124.62(Ph-C, 2C), 114.74(Ph-C, 2C), 59.30(CH_2_), 55.70(OCH_3_). ESI-HRMS: calcd. for C_16_H_13_F_3_N_4_O_2_S [M + H]^+^, 383.0790; found, 383.0790.

5-((3-methoxybenzyl) sulfinyl)-2-(4-methoxyphenyl)-2*H*-tetrazole (**6k**): yellow oily liquid, yield 73%. ^1^H NMR (600 MHz, CDCl_3_) δ 7.22–7.18 (m, 1H, Ph-H), 7.13–7.09 (m, 2H, Ph-H), 6.96–6.92 (m, 2H, Ph-H), 6.88–6.85 (m, 1H, Ph-H), 6.76–6.72 (m, 1H, Ph-H), 6.66 (t, *J* = 2.0 Hz, 1H, Ph-H), 4.85–4.66 (m, 2H, CH_2_), 3.86 (d, *J* = 1.3 Hz, 3H, CH_3_), 3.71 (d, *J* = 1.2 Hz, 3H, CH_3_). ^13^C NMR (151 MHz, CDCl_3_) δ 161.41(Ph-C), 160.05(Ph-C), 156.19(N=C-N), 130.17(Ph-C), 128.69(Ph-C), 126.50(Ph-C, 2C), 125.35(Ph-C), 122.73(Ph-C), 115.48(Ph-C), 115.37(Ph-C), 114.66(Ph-C, 2C), 60.31(CH_2_), 55.70(OCH_3_), 55.30(OCH_3_). ESI-HRMS: calcd. for C_16_H_16_N_4_O_3_S [M + H]^+^, 345.1016; found, 345.1000.

5-((4-methoxybenzyl) sulfinyl)-2-(4-methoxyphenyl)-2*H*-tetrazole (**6l**): yellow solid, yield 83%. ^1^H NMR (600 MHz, CDCl_3_) δ 7.16–7.11 (m, 2H, Ph-H), 7.10–7.05 (m, 2H, Ph-H), 6.97–6.91 (m, 2H, Ph-H), 6.82–6.79 (m, 2H, Ph-H), 4.84–4.64 (m, 2H, CH_2_), 3.86 (s, 3H, CH_3_), 3.77 (s, 3H, CH_3_). ^13^C NMR (151 MHz, CDCl_3_) δ 161.38(Ph-C), 160.39(Ph-C), 156.16(N=C-N), 131.89(Ph-C, 2C), 126.46(Ph-C, 2C), 125.43(Ph-C), 119.00(Ph-C), 114.63(Ph-C, 2C), 114.58(Ph-C, 2C), 59.62(CH_2_), 55.70(OCH_3_), 55.35(OCH_3_). ESI-HRMS: calcd. for C_16_H_16_N_4_O_3_S [M + Na]^+^, 367.0835; found 367.0825.

5-((3, 5-dimethoxybenzyl) sulfinyl)-2-(4-methoxyphenyl)-2*H*-tetrazole (**6m**): yellow solid, yield 78%. ^1^H NMR (600 MHz, CDCl_3_) δ 7.18–7.13 (m, 2H, Ph-H), 6.95 (d, *J* = 8.9 Hz, 2H, Ph-H), 6.42–6.25 (m, 3H, Ph-H), 4.79–4.63 (m, 2H, CH_2_), 3.86 (s, 3H, CH_3_), 3.69 (s, 6H, CH_3_). ^13^C NMR (151 MHz, CDCl_3_) δ 161.40(Ph-C), 161.20(Ph-C), 156.34(N=C-N), 129.27(Ph-C), 126.46(Ph-C, 2C), 125.36(Ph-C), 114.67(Ph-C, 2C), 108.12(Ph-C, 2C), 101.59(Ph-C), 60.59(CH_2_), 55.71(OCH_3_), 55.42(OCH_3_, 2C). ESI-HRMS: calcd. for C_17_H_18_N_4_O_4_S [M + Na]^+^, 397.0911; found, 397.0941.

Methyl 2-((2-(4-methoxyphenyl)-2*H*-tetrazol-5-yl) sulfinyl) acetate (**6n**): oily liquid, yield 70%. ^1^H NMR (600 MHz, CDCl_3_) δ 7.63–7.60 (m, 2H, Ph-H), 7.11–7.08 (m, 2H, Ph-H), 4.89–4.48 (m Hz, 2H, CH_2_), 3.91 (s, 3H, CH_3_), 3.76 (s, 3H, CH_3_). ^13^C NMR (151 MHz, CDCl_3_) δ 164.92(Ph-C), 161.73(Ph-C), 156.29(N=C-N), 126.57(Ph-C, 2C), 125.36(Ph-C), 115.13(Ph-C, 2C), 56.27(CH_2_), 55.78(OCH_3_), 53.28(OCH_3_). ESI-HRMS: calcd. for C_11_H_12_N_4_O_4_S [M + Na]^+^, 319.0471; found 319.0467.

Ethyl 2-((2-(4-methoxyphenyl)-2*H*-tetrazol-5-yl) sulfinyl) acetate (**6o**): oily liquid, yield 77%. ^1^H NMR (600 MHz, CDCl_3_) δ 7.65–7.59 (m, 2H, Ph-H), 7.12–7.08 (m, 2H, Ph-H), 4.88–4.46 (m, 2H, CH_2_), 4.20 (qd, *J* = 7.2, 2.1 Hz, 2H, CH_2_), 3.91 (s, 3H, CH_3_), 1.26 (t, *J* = 7.2 Hz, 3H, CH_3_). ^13^C NMR (151 MHz, CDCl_3_) δ 164.44(Ph-C), 161.71(Ph-C), 156.36(N=C-N), 126.55(Ph-C, 2C), 125.39(Ph-C), 115.13(Ph-C, 2C), 62.75(CH_2_), 56.45(OCH_3_), 55.78(OCH_3_), 13.96(CH_3_). ESI-HRMS: calcd. for C_12_H_14_N_4_O_4_S [M + Na]^+^, 333.0628; found 333.0625.

2-(benzylsulfinyl)-5-(4-chlorophenyl)-1,3,4-oxadiazole (**7a**): white solid, yield 82%. ^1^H NMR (600 MHz, CDCl_3_) δ 7.99–7.96 (m, 2H, Ph-H), 7.52–7.48 (m, 2H, Ph-H), 7.35 (dd, *J* = 4.8, 1.9 Hz, 3H, Ph-H), 7.30 (dt, *J* = 4.7, 3.5 Hz, 2H, Ph-H), 4.71–4.58 (m, 2H, CH_2_). ^13^C NMR (151 MHz, CDCl_3_) δ 166.35(Ph-C), 165.60(Ph-C), 139.30(Ph-C), 130.36(Ph-C, 2C), 129.70(Ph-C, 2C), 129.29(Ph-C), 129.17(Ph-C, 2C), 128.69(Ph-C, 2C), 127.80(Ph-C), 121.01(Ph-C), 60.56(CH_2_). ESI-HRMS: calcd. for C_15_H_11_ClN_2_O_2_S [M + Na]^+^, 341.0122; found, 341.0103.

2-((4-chlorobenzyl) sulfinyl)-5-(4-chlorophenyl)-1,3,4-oxadiazole (**7b**): white solid, yield 82%. ^1^H NMR (600 MHz, CDCl_3_) δ 7.97 (dd, J = 8.3, 6.4 Hz, 2H, Ph-H), 7.54–7.49 (m, 2H, Ph-H), 7.35–7.30 (m, 2H, Ph-H), 7.26–7.24 (m, 2H, Ph-H), 4.70–4.52 (m, 2H, CH_2_). ^13^C NMR (151 MHz, CDCl_3_) δ 166.50(C=N), 165.34(C=N), 139.42(Ph-C), 135.65(Ph-C), 131.73(Ph-C, 2C), 130.92(Ph-C), 129.75(Ph-C, 2C), 129.38(Ph-C), 128.68(Ph-C, 2C), 126.30(Ph-C), 120.90(Ph-C), 59.53(CH_2_). ESI-HRMS: calcd. for C_15_H_10_Cl_2_N_2_O_2_S [M + Na]^+^, 374.9732; found, 374.9717.

2-(4-chlorophenyl)-5-((naphthalen-2-ylmethyl) sulfinyl)-1,3,4-oxadiazole (**7f**): white solid, yield 86%. ^1^H NMR (600 MHz, CDCl_3_) δ 7.88–7.83 (m, 2H, Ph-H), 7.82–7.77 (m, 4H, Ph-H), 7.51–7.46 (m, 2H, Ph-H), 7.45–7.43 (m, 2H, Ph-H), 7.35–7.33 (m, 1H, Ph-H), 4.89–4.66 (m, 2H, CH_2_). ^13^C NMR (151 MHz, CDCl_3_) δ 166.35(C=N), 165.62(C=N), 139.20(Ph-C), 133.32(Ph-C), 133.21(Ph-C), 130.24(Ph-C), 129.60(Ph-C, 2C), 129.03(Ph-C), 128.61(Ph-C, 2C), 128.00(Ph-C), 127.72(Ph-C), 127.09(Ph-C), 126.94(Ph-C), 126.74(Ph-C), 125.14(Ph-C), 120.90(Ph-C), 60.88(CH_2_). ESI-HRMS: calcd. for C_19_H_13_ClN_2_O_2_S [M + Na]^+^, 391.0278; found, 391.0259.

2-(4-chlorophenyl)-5-((4-nitrobenzyl) sulfinyl) -1,3,4-oxadiazole (**7g**): yellow solid, yield 81%. ^1^H NMR (600 MHz, CDCl_3_) δ 8.31–8.18 (m, 2H, Ph-H), 8.02–7.99 (m, 2H, Ph-H), 7.59–7.52 (m, 4H, Ph-H), 4.88–4.65 (m, 2H, CH_2_). ^13^C NMR (151 MHz, CDCl_3_) δ 166.76(C=N), 164.95(C=N), 148.47(Ph-C), 139.64(Ph-C), 135.01(Ph-C), 131.63(Ph-C, 2C), 129.82(Ph-C, 2C), 128.68(Ph-C, 2C), 124.11(Ph-C, 2C), 120.73(Ph-C), 58.98(CH_2_). ESI-HRMS: calcd. for C_15_H_10_ClN_3_O_4_S [M + Na]^+^, 385.9973; found, 385.9965.

2-(4-chlorophenyl)-5-((4-fluorobenzyl) sulfinyl)-1,3,4-oxadiazole (**7h**): white solid, yield 87%. ^1^H NMR (600 MHz, CDCl_3_) δ 8.01–7.97 (m, 2H, Ph-H), 7.54–7.50 (m, 2H, Ph-H), 7.33–7.28 (m, 2H, Ph-H), 7.08–7.01 (m, 2H, Ph-H), 4.77–4.41 (m, 2H, CH_2_). ^13^C NMR (151 MHz, CDCl_3_) δ 166.45(C=N), 165.42(C=N), 164.15(Ph-C), 162.50(Ph-C), 139.39(Ph-C), 132.25(Ph-C), 129.73(Ph-C, 2C), 128.66(Ph-C, 2C), 123.65(Ph-C), 120.91(Ph-C), 116.33(Ph-C), 116.18(Ph-C), 59.42(CH_2_). ESI-HRMS: calcd. for C_15_H_10_ClFN_2_O_2_S [M + Na]^+^, 359.0028; found, 359.0009.

2-((4-bromobenzyl) sulfinyl)-5-(4-chlorophenyl)-1,3,4-oxadiazole (**7i**): white solid, yield 79%. ^1^H NMR (600 MHz, CDCl_3_) δ 8.00–7.94 (m, 2H, Ph-H), 7.54–7.51 (m, 2H, Ph-H), 7.50–7.47 (m, 2H, Ph-H), 7.20–7.17 (m, 2H, Ph-H), 4.68–4.52 (m, 2H, CH_2_). ^13^C NMR (151 MHz, CDCl_3_) δ 166.51(C=N), 165.31(C=N), 139.42(Ph-C), 132.34(Ph-C, 2C), 131.99(Ph-C, 2C), 129.76(Ph-C, 2C), 128.69(Ph-C, 2C), 126.81(Ph-C), 123.86(Ph-C), 120.88(Ph-C), 59.61(CH_2_). ESI- HRMS: calcd. for C_15_H_10_BrClN_2_O_2_S [M + Na]^+^, 418.9227; found 418.9229.

2-(4-chlorophenyl)-5-((4-methoxybenzyl) sulfinyl)-1,3,4-oxadiazole (**7l**): white solid, yield 68%. ^1^H NMR (600 MHz, CDCl_3_) δ 8.00–7.93 (m, 2H, Ph-H), 7.55–7.49 (m, 2H, Ph-H), 7.18–7.12 (m, 4H, Ph-H), 4.68–4.50 (m, 2H, CH_2_), 2.31 (s, 3H, CH_3_). ^13^C NMR (151 MHz, CDCl_3_) δ 166.29(C=N), 165.66(C=N), 139.34(Ph-C), 139.27(Ph-C), 130.21(Ph-C, 2C), 129.87(Ph-C, 2C), 129.67(Ph-C, 2C), 128.68(Ph-C, 2C), 124.57(Ph-C), 121.06(Ph-C), 60.35(CH_2_), 21.19(CH_3_). ESI-HRMS: calcd. for C_16_H_13_ClN_2_O_3_S [M + Na]^+^, 371.0228; found, 371.0238.

2-(4-chlorophenyl)-5-((3,5-dimethoxybenzyl) sulfinyl)-1,3,4-oxadiazole (**7m**): yellow solid, yield 71%. ^1^H NMR (600 MHz, CDCl_3_) δ 8.03–7.97 (m, 2H, Ph-H), 7.55–7.49 (m, 2H, Ph-H), 6.42 (s, 3H, Ph-H), 4.63–4.54 (m, 2H, CH_2_), 3.72 (s, 6H, CH_3_). ^13^C NMR (151 MHz, CDCl_3_) δ 166.35(C=N), 165.76(C=N), 161.23(Ph-C), 139.33(Ph-C), 129.75(Ph-C), 129.72(Ph-C, 2C), 128.69(Ph-C, 2C), 126.96(Ph-C), 121.02(Ph-C), 108.13(Ph-C, 2C), 101.35(Ph-C), 60.99(CH_2_), 55.39(OCH_3_, 2C). ESI-HRMS: calcd. for C_17_H_15_ClN_2_O_4_S [M + Na]^+^, 401.0333; found, 401.0324.

Methyl 2-((5-(4-chlorophenyl)-1,3,4-oxadiazol-2-yl) sulfinyl) acetate (**7n**): white solid, yield 88%. ^1^H NMR (600 MHz, CDCl_3_) δ 8.15–8.03 (m, 2H, Ph-H), 7.59–7.51 (m, 2H, Ph-H), 4.71–4.32 (m, 2H, CH_2_), 3.86 (s, 3H, CH_3_). ^13^C NMR (151 MHz, CDCl_3_) δ 166.76(C=N), 165.54(C=N), 164.25(C=O), 139.55(Ph-C), 129.80(Ph-C, 2C), 128.82(Ph-C, 2C), 120.94(Ph-C), 57.14(CH_2_), 53.48(OCH_3_). ESI-HRMS: calcd. for C_11_H_9_ClN_2_O_4_S [M + Na]^+^, 322.9864; found, 322.9850.

Ethyl 2-((5-(4-chlorophenyl)-1,3,4-oxadiazol-2-yl) sulfinyl) acetate (**7o**): yellow solid, yield 81%. ^1^H NMR (600 MHz, CDCl_3_) δ 8.09–8.06 (m, 2H, Ph-H), 7.56–7.53 (m, 2H, Ph-H), 4.55–4.38 (m, 2H, CH_2_), 4.28 (q, *J* = 7.1 Hz, 2H, CH_2_), 1.29 (t, *J* = 7.1 Hz, 3H, CH_3_). ^13^C NMR (151 MHz, CDCl_3_) δ 166.68(C=N), 165.64(C=N), 163.73(C=O), 139.52(Ph-C), 129.79(Ph-C, 2C), 128.80(Ph-C, 2C), 120.97(Ph-C), 62.95(CH_2_), 57.38(OCH_2_), 14.00(CH_3_). ESI-HRMS: calcd. for C_12_H_11_ClN_2_O_4_S [M + Na]^+^, 337.0020; found, 337.0016.

### 3.2. Biological Investigations

#### 3.2.1. Biofilm Formation Assay

The biofilm was quantitatively analyzed by the crystal violet method [44]. The log phase of *P. aeruginosa* PAO1 was diluted 1000-fold with fresh Luria broth (LB) medium, then the diluted bacterial solution and the compound solution were added to a 96-well plate, making the final concentration of the compounds 50 μM. The control group was added to 0.5% DMSO per well. The 96-well plates were incubated at 37 °C for 24 h, then OD of the suspended cells was measured at 600 nm using a microplate reader (Molecular Devices, Spectra Max M5, Sunnyvale, CA, USA). The bacterial culture was removed and washed three times with PBS and dried; 0.1% crystal violet solution was then added to each well for 15 min staining. After the excess crystal violet solution was removed and washed three times with PBS, the pigment was dissolved in 95% ethanol. Finally, the absorbance value was measured at 595 nm by a microplate reader. The formula for calculating the biofilm inhibition rate is: OD_595control_ − OD_595_/OD595_control_ × 100%.

#### 3.2.2. *P. aeruginosa* QS Inhibition Assays [32,45]

Test compounds were dissolved in 100% DMSO to a concentration of 10 mM and mixed with ABTGC medium, and they were then added to 96-well microtiter plates (Corning, Corning, NY, USA,) in the first well, giving a final concentration of 40 μM in a volume of 100 μL. An overnight culture of the PAO1-*lasB*-*gfp* strain, which had grown in LB medium at 37 °C with 200 rpm, was diluted in ABTGC medium to an optical density at 600 nm (OD_600_) of 0.2. Then 100 μL bacterial suspension was added to the wells of the microtiter plate to reach a final inhibitor concentration of 20 μM. DMSO control with 0.2% final concentration was used. The microtiter plate was incubated in a Molecular Devices SpectraMax microplate reader at 37 °C, with GFP fluorescence signals (excitation 485 nm, emission 535 nm) and cell density (OD_600_) measured every 20 min for at least 12 h. The Inhibition assay of all test compounds and controls were determined in triplicate. *P. aeruginosa* Rhl and Pqs inhibition assays were performed using a similar method to that of the LasB inhibition assay.

#### 3.2.3. CLSM Images

*P. aeruginosa* PAO1 was cultured overnight and diluted 100-fold, then the test compounds and bacterial suspension were added to the plate, followed by incubation at 37 °C for 24 h. Floating bacteria were poured out and washed with water three times, then fixed with 4% paraformaldehyde for 15 min and stained with 0.01% acridine orange for 15 min in the dark; excess dye was then washed with PBS. The established model was observed by confocal laser scanning microscope (Nikon-Eclipse-Ti) under green fluorescence light (excitation wavelength: 488 nm, emission wavelength: 515 nm). The signal was received by the FITC channel, where the objective lens was ×10, and scanned layer by layer along the Z-axis from outside to inside.

#### 3.2.4. Quantification Analysis of Elastase

The elastase quantification assay was performed as previously described [46]. An overnight culture of *P. aeruginosa* PAO1 was diluted to a density of OD_600_ = 0.01 in LB medium and inoculated with compounds in a 20 mL conical flask, then incubated at 37 °C with shaking at 200 rpm for 24 h. The cultures were centrifuged at 10,000 rpm at 4 °C for 10 min, and the supernatant was collected and filtered with a 0.22 mm-pore size filter. A supernatant fraction of 100 µL was added to 900 µL of Elastin-Congo Red reaction buffer (2 mg/mL ECR, 0.1 mM Tris-HCl), which was shaken at 37 °C for 18 h. The reaction was placed on ice and 100 μL of 0.12 M EDTA was added to terminate the reaction, before being centrifuged at 4 °C, 12,000 r/min for 10 min; the supernatant was measured at 495 nm.

#### 3.2.5. Determination of Pyocyanin Production

The pyocyanin quantification assay is based on the absorbance of pyocyanin at 520 nm in acidic solution [47]. *P. aeruginosa* PAO1 was cultured overnight and diluted to OD_600_ = 0.01 in LB medium, and then inoculated with compounds in a 20 mL conical flask before being incubated at 37 °C, with shaking at 200 rpm for 24 h. The bacterial culture was centrifuged at 10,000 rpm for 10 min; the supernatant was collected and extracted with chloroform, then to the chloroform layer was added 0.2 M HCl for extraction (after the hydrochloric acid mixed reaction turned pink). The absorbance of the HCl layer was measured at 520 nm by microplate reader.

#### 3.2.6. Detection of Rhamnolipid Production

Rhamnolipid production was directly quantified using the orcinol assay according to the original protocol by Koch et al. [48]. *P. aeruginosa* PAO1 overnight culture was diluted 100-fold in LB medium and inoculated with compounds in a 20 mL conical flask. The cultures were incubated for 24 h at 37 °C, under shaking condition (200 rpm). Supernatants were collected after the mixture was centrifuged at 10,000 rpm for 10 min and extracted twice by diethyl ether. The ether fraction was evaporated to dry and then resuspended in deionized water and supplemented with orcinal solution (0.19% (*w*/*v*) orcinol, 50% H_2_SO_4_). The mixture was incubated in a water bath at 80 °C for 30 min, and then cooled at room temperature for 15 min; the OD value at 421 nm was measured.

### 3.3. Molecular Docking

Compounds and OdDHL were drawn with ChemBioDraw Ultra (ver. 13.0) software (Cambridge, MA, USA) and minimized with Molecular Operate Environment software (Innovation Center of Pesticide Research, Department of Applied Chemistry, College of Science, China Agricultural University, Beijing, China). The receptor protein lasR in PDB format was downloaded from the RCSB Protein Data Bank (http://www.pdb.org (accessed on 20 July 2022)). A LasR X-ray crystal structure with 1.80 A° resolution (PDB ID: 2UV0) was used for the docking study [49]. The process of deleting water, adding hydrogen, adding Gasteiger charges, and so on, was prepared by Autodock Tools software (San Diego, CA, USA). The OdDHL binding pocket was selected as the docking site, and the docking environment was set in the solvent. Docking was performed after the setting method (placement: triangular matcher, refinement: rigid receptor), score (placement: London dG, refinement: GBVI/WSA dG), and posture (placement: 30, refinement: 5). The optimal docking posture was selected to analyze the interaction between LasR and the target compounds.

## 4. Conclusions

The establishment of biofilms can protect *P. aeruginosa* and help to elude eradication by human immunity and antibacterial drugs. Clinically, once the biofilm is formed on abiotic or tissue surfaces, it is generally considered a pathogenic characteristic of chronic infection. The difficulty in treating *P. aeruginosa* infections with antibiotics is that almost all patients with cystic fibrosis eventually contract an incurable resistant strain [3,22]. QSIs that reduce the virulence of pathogens without killing pathogenic bacteria are considered feasible targets for the development of antimicrobial agents against *P. aeruginosa*. They can also alleviate the pressure of drug resistance to a certain extent. In this study, phenyloxadiazole sulfoxide derivative **5b** could inhibit the biofilm formation of *P. aeruginosa* but had no growth inhibitory effect. Similarly, in phenotypic experiments with virulence factors, the decreased production of extracellular virulence factor elastase was observed. Mechanism research results confirm that **5b** can effectively inhibit the *las* system in a dose-dependent manner, and the IC_50_ value of inhibitory concentration against the PAO1-*lasB*-*gfp* strain was 3.53 ± 0.16 μM. Molecular docking analysis showed that **5b** and LasR receptor proteins inhibited the production of virulence factors and *Pseudomonas aeruginosa* biofilms by forming hydrogen bonds and hydrophobic interactions. In conclusion, sulfoxide derivatives have been proposed as a new QSIs model, which provides a new strategy for the development of new antimicrobial agents and attenuating the pathogenicity of *P. aeruginosa*.

## Data Availability

The data presented in this study are available in the Appendix A.

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
