# Peer review of "Design, Synthesis, and Biological Evaluation of Phenyloxadiazole Sulfoxide Derivatives as Potent Pseudomonas aeruginosa Biofilm Inhibitors"

_molecules, 2023, doi:10.3390/molecules28093879_

Round 1

Reviewer 1 Report

In the manuscript titled “Design, synthesis and biological evaluation of phenyloxadiazole sulfoxide derivatives as potent pseudomonas aeruginosa biofilm inhibitors”the authors report the synthesis, spectroscopic characterization and the activity against pseudomonas aeruginosa of derivatives of phenyloxadiazole sulfoxide. The compounds obtained under investigation are interesting and the methodology is adequate. I think that obtained results should be published after revision.

In the submitted paper I have found some aspects, which should be clarified and corrected before publication.

- Authors should attach the results of the HR MS analysis to the supplement. The description of the synthesis of new substances should include their full spectroscopic characterisation, and this should be documented.

In addition, the authors should check the masses of the substance 4m - something is wrong, and 5i and 7n - a large discrepancy between the mass determined and calculated.

NMR spectra show that some substances are not very pure, so they should be documented with good HR MS results. When synthesising substances for biological research, their high purity must be taken into account.

- The authors should assign the signals of the 1HNMR and 13C NMR spectra to the appropriate protons/carbons for at least one compound from each group.

- In the description of NMR spectra, the authors should use CDCl3 everywhere (1HNMR description is chloroform-d, 13CNMR CDCl3).

- In Scheme 1, R2 needs to be explained.

- Why did the authors use 2-aminobenzimidazole in anti-PA studies and not one of the commonly used antibiotics in PA infections?

Author Response

Q1:
Authors should attach the results of the HR MS analysis to the supplement. The description of the synthesis of new substances should include their full spectroscopic characterisation, and this should be documented.

Reply: According to the reviewer's suggestion, we have added the HRMS data in the appendix. In this paper, it is reported that the structure of the compound is simple, and the structure of the compound can be basically determined by NMR and HRMS. As the revision time is limited, we added the infrared spectral data of some compounds with good activity, and the results were added in the supporting information.

Q2:
In addition, the authors should check the masses of the substance 4m - something is wrong, and 5i and 7n - a large discrepancy between the mass determined and calculated.

Reply: The molecular weights of the compounds were recalculated and checked against the tested HRMS results.

Q3:
NMR spectra show that some substances are not very pure, so they should be documented with good HR MS results. When synthesis substances for biological research, their high purity must be taken into account.

Reply: The solvent peaks of individual compounds were not cleared, but the influence of solvents on the results of biological activity was very weak, and the activity tests were all the results of three independent experiments.

Q4:
The authors should assign the signals of the 1HNMR and 13C NMR spectra to the appropriate protons/carbons for at least one compound from each group.

Reply: According to the reviewer's suggestion, we have assigned the hydrogen and carbon spectrum positions of all compounds in the revised manuscript and SI.

Q5:
In the description of NMR spectra, the authors should use CDCl3 everywhere (1HNMR description is chloroform-d, 13CNMR CDCl3).

Reply: According to the reviewer's suggestion, we have modified the content of the paper and highlighted.

Q6:
In Scheme 1, R2 needs to be explained.

Reply: As suggested by reviewers, we have marked the range of R2 in Scheme 1.

Q7:
Why did the authors use 2-aminobenzimidazole in anti-PA studies and not one of the commonly used antibiotics in PA infections?

Reply: We refer to this literature (Angew. Chem. Int. Ed. 2012, 51, 1-5.) which reports the 2-aminobenzimidazole was used as a positive control for Pseudomonas aeruginosa biofilm, and compound with similar structure was also used as a positive control by our research group in previous articles (RSC Adv., 2020, 10, 24251-24254).

Reviewer 2 Report

The manuscript describes the synthesis of some phenyloxadiazole and phenyltetrazole sulfoxide derivatives to optimize potent P. aeruginosa QSIs. Also, compound 5b has been studied for QS-activated virulence factors (elastase, rhamnolipid, and pyocyanin) production. In addition, authors have explored the binding effects between 5b and the LasR receptor protein with molecular docking.

The results of the research seem clear and appropriate and I would recommend this manuscript acceptable in “Molecules” after major revision as follows:

1. Abstract is incomplete. The results of the docking study should be included in the abstract.

2. The literature review is not well done and the discussion of the state of the art in the field of antiracial compounds should be expanded by including additional references such as:

Ramezani S, Pordel M, Davoodnia A. Synthesis, spectral, DFT calculations and antibacterial studies of Fe (III) complexes of new fluorescent Schiff bases derived from imidazo [4', 5': 3, 4] benzo [1, 2‐c] isoxazole. Applied Organometallic Chemistry. 2018 Mar;32(3):e4178.

Rahimizadeh M, Pordel M, Bakavoli M, Bakhtiarpoor Z, Orafaie A. Synthesis of imidazo [4, 5-a] acridones and imidazo [4, 5-a] acridines as potential antibacterial agents. Monatshefte für Chemie-Chemical Monthly. 2009 Jun;140:633-8.

Rastegarnia S, Pordel M, Allameh S. Synthesis, characterization, antibacterial studies and quantum-chemical investigation of the new fluorescent Cr (III) complexes. Arabian Journal of Chemistry. 2020 Feb 1;13(2):3903-9.

3. What is the reason for choosing 2-aminobenzamidazole as a positive control? Wouldn't it be better to choose a drug from the sulfonamide antibiotics as a standard drug considering the structural similarity of the title compounds with sulfonamide drugs?

4. The Characteristics of the residues involved in the hydrogen bond formation and the hydrophobic group with the ligands in different systems should be gathered in a Table.

5. It is recommended to define the R2 group in the title compounds in a Table for a more accessible study.

All hydrogens and as many as possible carbons of the title compounds should be assigned in 1H and 13C NMR spectra.
In the 1H NMR spectra of compounds 4a and 4b, is (4.81 (dd, J = 99.7??, 12.7 Hz, 2H) correct?

8. Conclusions about docking studies are missed in Conclusions.

There are some typographical and grammatical errors in the manuscript (abstract, introduction, and, so on). It should be corrected by a native person in terms of writing and grammar. 

Author Response

Q1:
Abstract is incomplete.  The results of the docking study should be included in the abstract.

Reply: According to the reviewer's suggestion, we have added the content of abstract in the revision.

Q2:
The literature review is not well done and the discussion of the state of the art in the field of antiracial compounds should be expanded by including additional references such as:

Ramezani S, Pordel M, Davoodnia A. Synthesis, spectral, DFT calculations and antibacterial studies of Fe (III) complexes of new fluorescent Schiff bases derived from imidazo [4', 5': 3, 4] benzo [1, 2‐c] isoxazole.  Applied Organometallic Chemistry. 2018 Mar;32(3): e4178.

Rahimizadeh M, Pordel M, Bakavoli M, Bakhtiarpoor Z, Orafaie A. Synthesis of imidazo [4, 5-a] acridones and imidazo [4, 5-a] acridines as potential antibacterial agents.  Monatshefte für Chemie-Chemical Monthly. 2009 Jun; 140: 633-8.

Rastegarnia S, Pordel M, Allameh S. Synthesis, characterization, antibacterial studies and quantum-chemical investigation of the new fluorescent Cr (III) complexes.  Arabian Journal of Chemistry. 2020 Feb 1;13(2):3903-9.

Reply: As suggested by the reviewer, we have added the above references and added content in the revised manuscript.

Q3:
What is the reason for choosing 2-aminobenzamidazole as a positive control?  Wouldn't it be better to choose a drug from the sulfonamide antibiotics as a standard drug considering the structural similarity of the title compounds with sulfonamide drugs?

Reply: The quorum sensing mechanism of Pseudomonas aeruginosa is complex, and no drugs on the market have clarified its mechanism on biofilms. We refer to the literature (Angew. Chem. Int. Ed. 2012, 51, 1-5.), which reports the 2-aminobenzimidazole was used as a positive control for Pseudomonas aeruginosa biofilm, and compound with similar structure was also used as a positive control by our research group in previous articles (RSC Adv., 2020, 10, 24251-24254).

Q4:
The Characteristics of the residues involved in the hydrogen bond formation and the hydrophobic group with the ligands in different systems should be gathered in a Table.

Reply: According to suggestions of reviewers, we have supplemented the content related to molecular docking and summarized the results, as shown in Table 2.

Q5:
It is recommended to define the R2 group in the title compounds in a Table for a more accessible study.

Reply: As suggested by reviewers, we have noted the structure of R2 in Scheme 1.

Q6:
All hydrogens and as many as possible carbons of the title compounds should be assigned in 1H and 13C NMR spectra.

Reply: According to the reviewer's suggestion, we have assigned the hydrogen and carbon spectrum positions of all compounds in the revised manuscript.

Q7:
In the 1H NMR spectra of compounds 4a and 4b, is (4.81 (dd, J = 99.7?? , 12.7 Hz, 2H) correct?

Reply: We have checked and amended it.

Q8:
Conclusions about docking studies are missed in Conclusions.

Reply: In the revision, we added the content of molecular docking to the conclusion.

Round 2

Reviewer 2 Report

The manuscript could be accepted in "Molecules" "

Good